# Unified Reward Model for Multimodal Understanding and Generation

## Abstract

Recent advances in human preference alignment have significantly enhanced multimodal generation and understanding. A key approach is training reward models to guide preference optimization. However, existing reward models are often task-specific, limiting their adaptability across diverse visual applications. We also argue that a reward model jointly learning to assess multiple vision tasks may foster a synergistic effect, where improved image understanding enhances image generation assessment, and refined image evaluation benefits video assessment through better frame analysis. To this end, this paper proposes UNIFIEDREWARD, the first unified reward model for multimodal understanding and generation assessment, enabling both pairwise ranking and pointwise scoring, which can be employed for vision model preference alignment. Specifically, (1) we first develop UNIFIEDREWARD on our constructed large-scale human preference dataset, including both image and video generation/understanding tasks. (2) Then, it is utilized to automatically construct high-quality preference pair data based on the vision models, fine-gradually filtering their outputs through pair ranking and point sifting. (3) Finally, these data are used for their preference alignment through Direct Preference Optimization (DPO). Experimental results demonstrate that joint learning to assess diverse visual tasks can lead to substantial mutual benefits, and we apply our pipeline to both image and video understanding/generation tasks, significantly improving the performance in each domain.

## 1 Introduction

Recent advancements in human preference alignment have substantially propelled the progress of multimodal generation and understanding tasks. A straightforward technique is directly collecting human feedback to construct preference datasets for model optimization (Wallace et al., 2024; Liu et al., 2024a; Zhang et al., 2024b). Despite its effectiveness, collecting large-scale human feedback is time-consuming and resource-intensive. To this end, an alternative popular approach involves learning reward models (Wang et al., 2024b; Xiong et al., 2024; Zang et al., 2025; Liu et al., 2025; Xu et al., 2024; Lee et al., 2023; Li et al., 2025) from a limited amount of preference data and using the learned reward function to generate preference data based on the output of vision models. This synthetic preference data can then be leveraged for vision model preference alignment, significantly reducing the need for extensive human annotations.

Despite their progress, we posit two concerns: **(1)** current reward models are often tailored to specific tasks, as shown in Tab. 1, limiting their adaptability across diverse visual understanding and generative tasks. The key challenge lies in the lack of a comprehensive human preference dataset that spans a wide range of visual tasks. **(2)** We intuitively argue that visual tasks are inherently interconnected, and jointly learning multiple visual tasks may create a mutually reinforcing effect. Specifically, enhanced image understanding may improve the evaluation of image generation by providing a more accurate assessment of content quality and contextual relevance. Similarly, improvements in image evaluation may benefit video evaluation, as high-quality image assessments lead to more accurate evaluations of video frames, contributing to overall better quality video assessment. This cross-task synergy facilitates a more robust evaluation of outputs across both image and video modalities in

---

*Equal contribution. †Corresponding author.

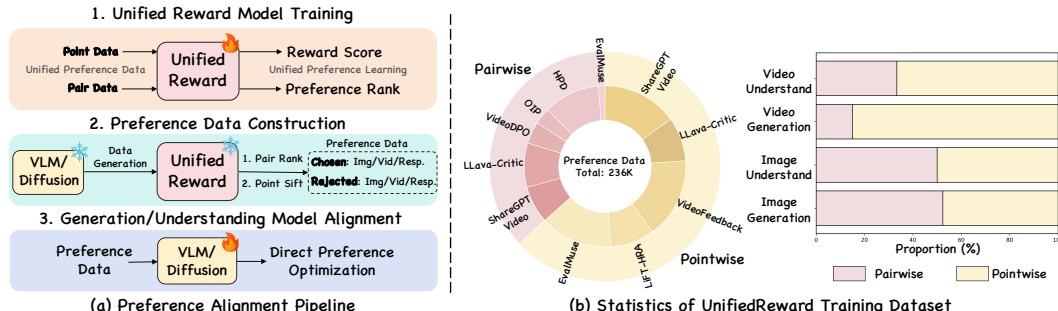

(a) Preference Alignment Pipeline      (b) Statistics of UnifiedReward Training Dataset

Figure 1: **Pipeline and Dataset Overview**. (a) Our proposed general preference alignment pipeline; (2) Statistics of UNIFIEDREWARD training dataset.

Table 1: **Comparison of Our Reward Model with Recent Approaches.** UNIFIEDREWARD is capable of assessing both image and video understanding and generation. "Pair" and "Point" refer to "Pair Ranking" and "Point Scoring", respectively.

| Reward Model | Method | Image Generation | Image Understand | Video Generation | Video Understand |
|---|---|---|---|---|---|
| PickScore'23 Kirstain et al. (2023) | Point | ✓ | | | |
| HPS'23 Wu et al. (2023) | Point | ✓ | | | |
| ImageReward'23 Wu et al. (2023) | Point | ✓ | | | |
| LLaVA-Critic'24 Xiong et al. (2024) | Pair/Point | | ✓ | | |
| VideoScore'24 Wang et al. (2024b) | Point | | | ✓ | |
| LiFT'24 Wang et al. (2024b) | Point | | | ✓ | |
| VisionReward'24 Xu et al. (2024) | Point | ✓ | | ✓ | |
| VideoReward'25 Liu et al. (2025) | Point | | | ✓ | |
| **UnifiedReward** | Pair/Point | ✓ | ✓ | ✓ | ✓ |

tasks involving understanding and generation. It inspires the development of a unified modal reward model that yields more precise reward signals for preference optimization.

To this end, we propose **UNIFIEDREWARD**, the first unified reward model for multimodal understanding and generation model assessment, capable of both pairwise ranking and pointwise scoring, which can be utilized for preference alignment on diverse vision tasks. As shown in Fig. 1 (a), our fine-tuning pipeline includes three key stages: **(1)** First, we construct a large-scale human preference dataset that spans both image and video generation/understanding tasks and develop UNIFIEDREWARD based on this dataset. **(2)** Next, we employ UNIFIEDREWARD to automatically construct high-quality preference pair data by selecting the outputs of specific baselines, such as Vision Language Models (VLM) and diffusion models, through multi-stage filtering, *i.e.,* pair ranking and point sifting. **(3)** Finally, we use these preference pairs to align the outputs of these models with human preferences via direct preference optimization. Our experiments show that learning multiple visual tasks together yields significant reciprocal benefits, enhancing performance in each individual domain. By implementing our pipeline across both vision understanding and generation baselines, we observe notable improvements in each domain.

**Contributions**: (1) We construct a large-scale human preference dataset that spans diverse vision tasks and develop UNIFIEDREWARD, the first unified reward model for multimodal understanding and generation model assessment. (2) We propose a general pipeline for both vision understanding and generation model preference alignment, which remains an underexplored area in current research. Extensive experiments demonstrate its effectiveness in improving the performance of vision models in each domain. (3) Our experiments reveal that learning to assess image and video tasks jointly leads to a synergistic improvement in performance across different visual domains. Through this work, we aim to expand the scope of reward models, making them more adaptable, generalizable, and effective across various visual applications.

## 2 RELATED WORK

**Reward Models** are crucial in aligning vision understanding and generation models with human preferences. Traditional studies (Xu et al., 2023; Zhang et al., 2024d; Liang et al., 2024) utilize human preference data to fine-tune CLIP, enabling them to better predict and align with human evaluations.

With the advent of VLMs (Achiam et al., 2023; Wang et al., 2024a), their robust ability to align visual and textual data makes them promising candidates for reward modeling. These models can be adapted into two main categories based on their capabilities: understanding assessment models (Xiong et al., 2024; Zang et al., 2025), which are designed exclusively for evaluating visual understanding tasks, and generation assessment models (Wang et al., 2024b; Xu et al., 2024; Liu et al., 2025; He et al., 2024), which focus on assessing visual synthesis quality.

However, these reward models are typically designed for specific tasks, as illustrated in Tab. 1, restricting their ability to adapt to diverse visual understanding and generative tasks. In this work, we propose the first unified reward model for both image and video understanding and generation assessment, which is more adaptable, generalizable, and effective across various visual applications.

**Preference Learning for VLM/Diffusion** is widely utilized to enhance their image and video understanding/generation performance. In video understanding, prior works have explored reinforcement learning with human feedback (Sun et al., 2023c) and AI-generated feedback (Ahn et al., 2024; Li et al., 2025; Zhang et al., 2024b) to refine reward models to enhance video LMMs. For image understanding, researchers (Ahn et al., 2024; Gunjal et al., 2024) investigate Direct Preference Optimization (DPO) as an alternative approach to preference modeling. Similar methods have been applied to image generation (Wallace et al., 2024; Lee et al., 2023; Xu et al., 2024) and video generation (Liu et al., 2024a; 2025; Wang et al., 2024b; Yuan et al., 2024; Zhang et al., 2024a), using reward models or human preference data to align pre-trained diffusion models.

However, these methods rely on task-specific reward models, and no unified reward model has been developed for preference learning across both image and video generation and understanding tasks. This limits the generalizability and efficiency of reward-based alignment. Our work investigates the effectiveness of joint learning to assess multiple visual tasks, demonstrating that cross-task synergy enhances the evaluation capabilities across each domain.

## 3 METHOD

### 3.1 OVERVIEW

This work aims to propose a unified reward model for vision model preference alignment. Existing studies typically develop specialized reward models for specific tasks as shown in Tab. 1, which restricts their adaptability across diverse visual applications. Furthermore, we intuitively argue that jointly learning multiple visual tasks can create a mutually reinforcing effect, yet this remains an underexplored area. To this end, this work proposes UNIFIEDREWARD, the first unified reward model for multimodal understanding and generation assessment, enabling both pair ranking and point scoring. It is then leveraged for Vision-Language Models (VLMs) and Diffusion model alignment, enabling more robust and adaptable preference learning across diverse visual tasks.

Our pipeline is illustrated in Fig. 2. Specifically, we first construct a large-scale, unified preference dataset (Sec. 3.2.1) and train our UNIFIEDREWARD model on this dataset (Sec. 3.2.2). Then, we curate preference datasets for VLMs and diffusion models by applying pair ranking and point sifting on their outputs (Sec. 3.3). These curated datasets are subsequently used for Direct Preference Optimization (DPO) (Sec. 3.4), effectively enhancing model alignment with human preferences.

### 3.2 UNIFIED REWARD MODEL TRAINING

#### 3.2.1 UNIFIED PREFERENCE DATASET CONSTRUCTION

A comprehensive human preference dataset that spans multiple vision-related tasks is essential for training a unified reward model. However, existing human feedback datasets, such as (Wang et al., 2024b; Liu et al., 2024a; Xiong et al., 2024), are typically designed for specific tasks, limiting their generalizability. Currently, there is no human preference dataset that comprehensively covers both visual understanding and generation tasks, highlighting the need for a more versatile dataset. To bridge this gap, we integrate existing datasets and preprocess them to construct the first large-scale unified human preference dataset, which consists of approximately 236K data covering both image and video understanding and generation tasks. The detailed statistics and visualized distributions

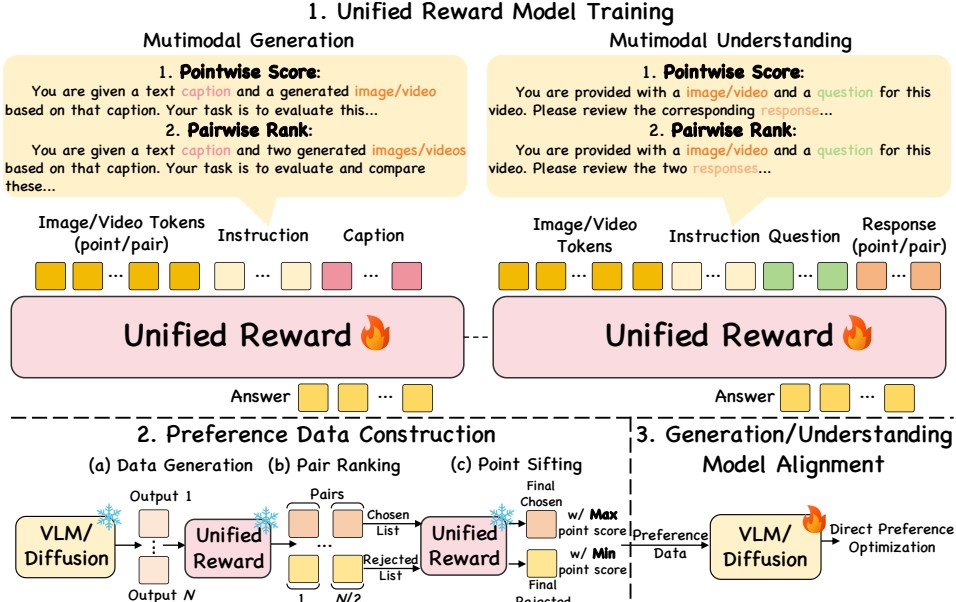

Figure 2: **Method Overview.** (1) *Unified Reward Model Training*: train a unified reward model for both multimodal generation and understanding assessment using pointwise scoring and pairwise ranking strategy. (2) *Preference Data Construction*: use the trained reward model to construct high-quality preference data through three steps: (a) data generation from vision models, (b) pairwise ranking to divide the chosen and rejected outputs, and (c) pointwise filtering to refine the chosen and rejected samples. (3) *Generation/Understanding Model Alignment*: the constructed preference data is then used to align vision models with human preference via Direct Preference Optimization.

of the dataset are presented in Fig. 1 (b) and Tab. 2, respectively. We will elaborate on the data construction process for each task in the following.

**Image Generation.** *EvalMuse* (Han et al., 2024) consists of 4K prompts, each with multiple images generated by different models. Each image is evaluated by at least three annotators, who provide an overall score (1-5) and element-wise labels indicating whether specific elements are present. For pointwise score learning, we compute the final score as the average of all ratings. An element is considered generated if at least two annotators agree; otherwise, it is marked as not generated. We integrate the overall score and element-wise labels as assessment answer for reward model learning. For pairwise ranking, we select the images with the highest and lowest average score from the same prompt as a ranking pair. *Human Preference Dataset (HPD)* (Christodoulou & Kuhlmann-Jørgensen, 2024) contains 700K human preference votes. For each prompt, two images generated by different models are provided, each with its respective vote count. In our work, we directly use the vote counts to construct pairwise ranking data, ranking the image with more votes as the preferred one. *Open-Image-Preferences (OIP)* [1] contains 7.4K text-to-image preference pairs, which are directly used in this work. **Image Understanding.** *LLava-Critic-113K* (Xiong et al., 2024) consists of 40K pointwise score and 73K pairwise ranking data samples for image understanding assessment learning. From this dataset, we select 25K samples each for pairwise ranking and pointwise scoring learning. **Video Generation.** *VideoDPO* (Liu et al., 2024a) includes 10K synthesized video pairs for text-to-video model DPO. We directly use this dataset for our pairwise ranking learning in video generation. *LiFT-HRA* (Wang et al., 2024b) and *VideoFeedback* (He et al., 2024) provide extensive human feedback for pointwise scoring of synthesized videos, which we directly incorporate into our work. **Video Understanding.** *ShareGPTVideo-DPO* (Zhang et al., 2024c) contains 17K video understanding DPO data, where each response in a pair is assigned an evaluation score. We directly use the pair data for pairwise ranking learning, while the individual response scores are extracted for pointwise scoring learning.

For pairwise ranking datasets, we standardize the answer format as "image/video/response X is better than image/video/response Y", where "X" and "Y" represent the assigned indices. If the dataset

---
[1] https://huggingface.co/datasets/data-is-better-together/open-image-preferences-v1-binarized

Table 2: **Training Datasets for Image and Video Generation/Understanding Assessment.** "*" indicates the dataset is preprocessed in our work.

|  | Task | Method | Dataset | Size |
|---|---|---|---|---|
| Image | Generation | Pair | EvalMuse* | 3K |
|  |  |  | HPD* | 25.6K |
|  |  |  | OIP | 7.4K |
|  |  | Point | EvalMuse* | 32.7K |
|  | Understanding | Pair | LLaVA-Critic | 25K |
|  |  | Point | LLaVA-Critic | 25K |
| Video | Generation | Pair | VideoDPO | 10K |
|  |  | Point | LiFT-HRA | 20K |
|  |  |  | VideoFeedback | 36.6K |
|  | Understanding | Pair | ShareGPTVideo | 17K |
|  |  | Point | ShareGPTVideo* | 34K |

includes evaluation justifications (Xiong et al., 2024; Wang et al., 2024b), we retain them to allow the model to learn from human reasoning. For pointwise scoring, we do not enforce a unified response format or score range, allowing the model to learn from diverse rating styles and scoring systems across different datasets. To ensure alignment between evaluation criteria and responses, we adjust instruction prompts accordingly. The prompting templates are provided in Appendix F.

As shown in Fig. 1, the training data for video generation pairwise ranking assessment is relatively limited compared to other tasks, but we believe that the synergistic effect of multitask learning can alleviate this deficiency. Overall, our dataset provides a diverse and comprehensive collection of human preferences, covering both pairwise ranking and pointwise scoring across image and video understanding/generation tasks. This enables effective reward model training, ensuring robust performance across multimodal understanding and generation applications.

### 3.2.2 UNIFIED PREFERENCE LEARNING

Based on the comprehensive datasets, we fine-tune a pre-trained VLM (Li et al., 2024c) with strong vision understanding capabilities to develop UNIFIEDREWARD, jointly training it across diverse vision tasks. Instead of learning evaluation from scratch, we integrate assessment ability as an additional discriminative skill, leveraging the model's existing visual comprehension to enhance its evaluation performance across various tasks.

Fig. 2 (top) illustrates our training process. Specifically, for multimodal generation evaluation, our model takes vision tokens, instruction input, and a caption as input. In contrast, for multimodal understanding, the caption is replaced by a question and the corresponding response(s), aligning the input format with the respective task requirements. The model is trained to predict the pointwise score or pairwise ranking based on the criteria specified in the instruction prompt. If the training data includes justifications, the model is also trained to generate detailed explanations to support its evaluations. During training, the optimization objective is standard cross-entropy loss, but it is computed only on the model's predicted answer.

After training our UNIFIEDREWARD, we leverage it for preference alignment in multimodal understanding and generation models. This process consists of two sequential steps: Preference Data Construction and Generation/Understanding Model Alignment. The following sections provide a detailed explanation of each step.

### 3.3 PREFERENCE DATA CONSTRUCTION

The quality of preference alignment data directly determines the effectiveness of model alignment. Existing methods (Wang et al., 2024b; Liu et al., 2025; Xiong et al., 2024) are often limited to a single evaluation strategy, either assigning pairwise rankings or pointwise scores to model outputs for preference data construction. In contrast, this work leverages both pairwise ranking and pointwise scoring capabilities of UNIFIEDREWARD, enabling a higher quality preference data construction pipeline, illustrated in Fig. 2 (bottom left).

Specifically, our pipeline includes three sequential steps: (1) **Data Generation**. Given an image/video-question pair (or generation prompt), a VLM (or diffusion model) generates multiple candidate outputs $\{O_1, O_2, \ldots, O_N\}$. These outputs serve as the initial pool for followed preference data filtering. (2) **Pair Ranking**. Given $N$ outputs, we group them into $N/2$ pairs and use our model to perform pairwise

ranking for each pair. Then, we classify these ranked pairs into a chosen list $\mathcal{C} = \{O_1^c, O_2^c, \ldots, O_{N/2}^c\}$ and a rejected list $\mathcal{R} = \{O_1^r, O_2^r, \ldots, O_{N/2}^r\}$. (3) **Point Sifting**. Finally, we apply our model to assign pointwise scores to all outputs in both the chosen list and the rejected list. The final preference data pair is determined as:

$$(O_c^* = \arg \max_{O \in \mathcal{C}} S(O), \quad O_r^* = \arg \min_{O \in \mathcal{R}} S(O)),$$

where $S(O)$ represents the pointwise score assigned by our model, $O_c^*$ is the most preferred output and $O_r^*$ is the least preferred output.

By combining pairwise ranking and pointwise scoring, the final preference data could provide a high-quality and reliable preference signal, effectively capturing both relative comparisons and absolute quality assessments.

### 3.4 GENERATION/UNDERSTANDING MODEL ALIGNMENT

After constructing the preference data, we leverage it for multimodal generation and understanding model alignment using DPO, which enables models to align their outputs with human preferences without explicit reward modeling, optimizing directly based on ranked preference pairs.

**DPO for Multimodal Generation**. For multimodal generation tasks, diffusion models (Ho et al., 2020) are widely used due to their strong capability in generating high-quality and diverse outputs across image and video synthesis. Therefore, we apply DPO on diffusion models to align their outputs with human preferences.

Given the constructed preference pair dataset $\mathcal{D}_{Gen} = \{(x_0^w, x_0^l)_i\}_{i=1}^M$, where $x_0^w$ and $x_i^l$ represents the preferred generated sample and the less preferred sample respectively, $M$ denotes the number of samples, we optimize the diffusion model by comparing the noise prediction differences between the fine-tuned model and a pre-trained reference model following (Wallace et al., 2024):

$$L(\theta) = -\mathbb{E}_{(x_0^w, x_0^l) \sim \mathcal{D}_{Gen}, t \sim \mathcal{U}(0,T), x_t^w \sim q(x_t^w | x_0^w), x_t^l \sim q(x_t^l | x_0^l)} \log \sigma \bigg( -\beta_g T \omega(\lambda_t)$$

$$\left( \|\epsilon^w - \epsilon_\theta(x_t^w, t)\|_2^2 - \|\epsilon^w - \epsilon_{\text{ref}}(x_t^w, t)\|_2^2 - \left( \|\epsilon^l - \epsilon_\theta(x_t^l, t)\|_2^2 - \|\epsilon^l - \epsilon_{\text{ref}}(x_t^l, t)\|_2^2 \right) \right) \bigg),$$

where $x_t^w$ and $x_t^l$ are the noisy latents derived from $x_0^w$ and $x_0^l$ at timestep $t$, respectively. $\epsilon_\theta(x_t^*, t)$ and $\epsilon_{\text{ref}}(x_t^*, t)$ denote the predicted noise from the fine-tuned and pre-trained reference diffusion models, respectively. $\beta_g$ is a temperature hyperparameter controlling optimization strength, $\sigma$ is the logistic function, $\lambda_t$ represents the signal-to-noise ratio, and $T\omega(\lambda_t)$ is a weighting function, which is treated as a constant equal to $\beta_g$ in this work.

This loss encourages the fine-tuned diffusion model to reduce the denoising error for preferred samples while increasing it for less preferred ones, thereby improving the generation quality.

**DPO for Multimodal Understanding**. Similar to generation alignment, we apply DPO to adjust the model's response preference for multimodal understanding models, *i.e.,* VLMs. Given an input $x$ (e.g., an image/video-question pair) with a preferred response $y_w$ and a less preferred response $y_l$ from preference pair dataset $\mathcal{D}_{\text{Und}}$, the optimization is formulated as:

$$\mathcal{L}(\theta) = -\mathbb{E}_{(x, y_w, y_l) \sim \mathcal{D}_{\text{Und}}} \left[ \beta_u \log \sigma \left( \log \frac{\pi_\theta(y_w|x)}{\pi_{\text{ref}}(y_w|x)} - \log \frac{\pi_\theta(y_l|x)}{\pi_{\text{ref}}(y_l|x)} \right) \right],$$

where $\pi_\theta(y_*|x)$ and $\pi_{\text{ref}}(y_*|x)$ are the response probability under the fine-tuned model and pre-trained reference model, respectively. $\beta_u$ is a hyperparameter that controls optimization sensitivity.

This loss encourages the fine-tuned VLMs to increase the likelihood of generating preferred responses while decreasing it for less preferred ones, thereby improving the model's alignment with human preferences and enhancing reasoning quality.

Table 3: **Image Understanding Assessment.** We evaluate various aspects on VLRewardBench.

| Models | General | Hallu. | Reason. | Overall Accuracy | Macro Accuracy |
|---|---|---|---|---|---|
| Gemini-1.5-Pro | 50.8 | 72.5 | 64.2 | **67.2** | 62.5 |
| GPT-4o | 49.1 | 67.6 | **70.5** | 65.8 | 62.4 |
| LLaVA-Critic | 47.4 | 38.5 | 53.8 | 46.9 | 46.6 |
| OV-7B | 32.2 | 20.1 | 57.1 | 29.6 | 36.5 |
| w/ Img. Und. | 47.6 | 38.3 | 54.5 | 47.4 | 46.8 |
| w/ Img. Und.+Gen. | 49.8 | 52.6 | 58.1 | 50.4 | 53.5 |
| w/ Img.+Vid. Und | 52.4 | 55.6 | 57.2 | 52.7 | 55.1 |
| **UnifiedReward** | **60.6** | **78.4** | 60.5 | **66.1** | **66.5** |

Table 4: **Video Understanding Assessment.** We evaluate the performance of our model using different training data configurations.

| | OV-7B | w/ Vid. Und. | w/ Vid.&Img. Und. | w/ Vid Und.&Gen. | **UnifiedReward** |
|---|---|---|---|---|---|
| **Acc.** | 48.2 | 74.2 | 76.6 | 78.6 | **84.0** |

**(a) Video Generation Comparison**

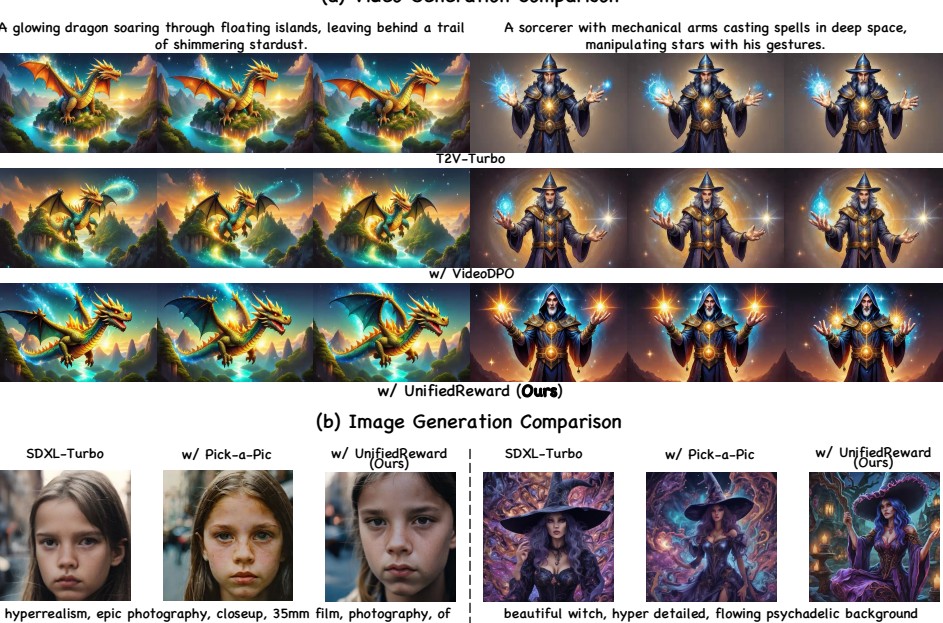

**(b) Image Generation Comparison**

Figure 3: **Qualitative Results**. (a) Video generation comparison. (b) Image generation comparison.

# 4 EXPERIMENTS

## 4.1 IMPLEMENTATION DETAILS

**Reward Model**: We adopt the pre-trained *LLaVA-OneVision* 7B (OV-7B) (Li et al., 2024c) as the base architecture for UNIFIEDREWARD. Training is conducted on 8 H100 GPUs with a batch size of 2, gradient accumulation steps of 16, a learning rate of $2.5 \times 10^{-6}$, and a warm-up ratio of 0.3. **Multimodal Understanding DPO**: Based on UNIFIEDREWARD, we apply DPO to *LLaVA-OneVision* 7B (Li et al., 2024c) and *LLaVA-Video* (Zhang et al., 2024e) to enhance their performance in image and video understanding, respectively. We use a batch size of 1, gradient accumulation steps of 16, a learning rate of $5 \times 10^{-7}$, and set $\beta_u = 0.1$. **Multimodal Generation DPO**: For image and video generation DPO, we use *SDXL-Turbo* (Podell et al., 2023) and *T2V-Turbo* (Li et al., 2024d), respectively. The parameter $\beta_g$ is set to 5000, with batch sizes of 32 for *SDXL-Turbo* and 16 for *T2V-Turbo*. We construct 10K preference data for video generation DPO and 14k for other task DPO. The number of candidate outputs $N$ is set to 10. All models are trained for 3 epochs. See Appendix A for details of baselines and benchmarks.

Table 5: **Image and Video Generation Comparison.** "tau" indicates that accuracy is calculated with ties, and "diff" excludes tied pairs when calculating accuracy.

| Method | Image Generation | | Method | Video Generation | | | |
| | GenAI-Bench | | | GenAI-Bench | | VideoGen-Reward | |
| | tau | diff | | tau | diff | tau | diff |
|---|---|---|---|---|---|---|---|
| PickScore | 53.2 | 67.2 | VideoScore | 46.2 | 70.6 | 42.1 | 49.9 |
| HPSv2 | 51.6 | 68.4 | LiFT | 41.2 | 60.1 | 40.6 | 58.3 |
| ImageReward | 47.8 | 65.0 | VisionReward | 52.1 | 73.1 | 57.4 | 68.2 |
| VisionReward | 46.8 | 66.4 | VideoReward | 50.2 | 73.3 | 60.1 | 73.9 |
| OV-7B | 39.7 | 53.2 | OV-7B | 40.8 | 51.4 | 40.4 | 50.2 |
| w/ Img. Gen. | 39.4 | 64.0 | w/ Vid. Gen. | 48.2 | 69.4 | 44.3 | 62.4 |
| w/ Img. Gen.+Und. | 47.7 | 65.9 | w/ Vid. Gen.+Und. | 49.1 | 71.6 | 45.1 | 64.9 |
| w/ Img.+Vid. Gen. | 50.5 | 67.6 | w/ Img.+Vid. Gen. | 52.0 | 73.6 | 53.6 | 70.7 |
| **UnifiedReward** | **54.8** | **70.9** | **UnifiedReward** | **60.7** | **77.2** | **66.6** | **79.3** |

Table 6: **Image Understanding DPO Comparison.** We compare our method with LLaVA-Critic for DPO based on LLaVA-OneVision-7B.

| | LLaVABen. | WildVision | LLaVABenWilder | LiveBen. | MMHal | MMBen | MME | MathVista | DocVQA | TextVQA |
|---|---|---|---|---|---|---|---|---|---|---|
| OV-7B | 90.3 | 54.9 | 67.8 | 77.1 | 3.19 | 80.9 | 1994.1 | 62.6 | 87.2 | **80.1** |
| w/ LLaVA-Critic'24 | 100.3 | 67.3 | 71.6 | 84.5 | 3.91 | 80.5 | 1998.9 | **63.2** | 86.98 | 79.2 |
| **w/ UnifiedReward** | **101.4** | **67.8** | **75.0** | **85.4** | **4.01** | **81.2** | **2008.5** | 62.9 | **87.4** | 79.5 |

## 4.2 REWARD MODEL COMPARISON RESULTS

**Image Understanding**. We compare our method with the latest open-source model, LLaVA-Critic (Xiong et al., 2024), as well as two closed-source models, Gemini-1.5-Pro (Team et al., 2024) and GPT-4o (Islam & Moushi, 2024). The experimental results, shown in Tab. 3, indicate that our method outperforms the best baseline in most metrics, e.g., macro accuracy (66.5% vs. 62.5%), which demonstrates the superiority of our method in image understanding assessment. For **Video Understanding**, to the best of our knowledge, there are currently no available baselines. Therefore, we explore the effectiveness of multi-task learning in video understanding assessment, which will be analyzed in the next section. In **Image Generation** assessment, we compare our method with both traditional and state-of-the-art approaches (Kirstain et al., 2023; Wu et al., 2023; Xu et al., 2023; 2024). The results are presented in Tab. 5. Notably, the latest work, VisionReward, supports reward modeling for both image and video generation. However, it trains separate models for each task using their respective datasets, whereas our approach jointly learns multiple tasks within a unified framework, leading to relatively better performance. For **Video Generation**, we compare our method with the latest approaches (He et al., 2024; Wang et al., 2024b; Xu et al., 2024; Liu et al., 2024a). As shown in Fig. 1, our training data for video generation assessment is relatively limited. However, as demonstrated in Tab. 5, our method excels across all metrics when compared to all baselines, highlighting that multitask learning not only mitigates the issue of insufficient training data but also enhances the learning effectiveness for video generation assessment.

### 4.2.1 MULTI-TASK ASSESSMENT LEARNING

This work intuitively argues that visual tasks are inherently interconnected, and jointly learning multiple visual tasks may create a mutually reinforcing effect. Therefore, we explore the effectiveness of multi-task learning on the reward model. Specifically, for each task, we employ different training data configurations to train the model, investigating the impact of jointly learning across different modalities (image and video) and tasks (understanding and generation). For example, for the image understanding task, we design three training configurations to investigate the impact of multi-task learning: (1) training solely on image understanding assessment, (2) jointly learning image understanding and image generation assessment, and (3) jointly learning image understanding and video understanding assessment. The results are presented in Tab. 3. Notably, our findings indicate that multi-task learning significantly enhances the model's overall performance compared to training on a single task. For instance, jointly training on both image and video understanding tasks improves overall accuracy and macro accuracy by 5.3% and 8.3%, respectively, compared to training solely on image understanding. Results for other tasks are presented in Tabs. 4 and 5, which consistently

Table 7: **Video Understanding DPO Comparison.** All methods are trained with same settings.

| Method | MSRVTT | | MSVD | | TGIF | | LongVideoBench | MLVU | Video-MME | | | |
| --- | --- | --- | --- | --- | --- | --- | --- | --- | --- | --- | --- | --- |
| | Acc. | Score | Acc. | Score | Acc. | Score | Acc. | M-Avg. | Short | Medium | Long | Avg. |
| LLaVA-Video-7B'24 | 52.8 | 3.24 | 69.7 | 3.90 | 51.9 | 3.37 | 58.1 | 70.9 | 76.1 | 61.6 | 52.3 | 63.3 |
| w/ Houd-DPO'24 | 56.8 | 3.34 | 72.8 | 3.97 | 54.9 | 3.45 | 58.0 | 71.8 | 76.3 | 61.3 | 51.2 | 63.0 |
| w/ TPO'25 | 55.0 | 3.25 | 72.6 | 3.93 | 53.7 | 3.40 | 58.2 | 72.6 | 76.9 | 62.1 | 52.1 | 63.7 |
| w/ **UnifiedReward** | 65.0 | 3.45 | 78.3 | 4.01 | 59.7 | 3.51 | 58.4 | 72.3 | 76.2 | 61.3 | 52.5 | 63.5 |

Table 8: **Image and Video Generation DPO Comparison.** We evaluate image generation using several image assessment models and evaluate video generation on VBench.

| Image Generation | | PickScore | HPSv2 | ImageReward |
| --- | --- | --- | --- | --- |
| SDXL-Turbo Podell et al. (2023) | Baseline | 43.24 | 29.37 | 0.82 |
| | w/ Pick-a-Pic | 54.32 | 30.03 | 0.93 |
| | w/ **UnifiedReward** | **63.32** | **32.44** | **1.05** |
| **Video Generation** | Models | VBench (%) | | |
| | | Total | Quality | Semantics |
| T2V-Turbo Li et al. (2024d) | Baseline | 80.95 | 82.71 | 73.93 |
| | w/ VideoDPO'24 | 81.80 | 83.80 | 73.81 |
| | w/ **UnifiedReward** | **82.10** | **84.11** | **74.06** |

demonstrate its effectiveness. These results highlight the benefits of leveraging shared knowledge across different visual tasks, leading to a more robust and generalizable reward model.

## 4.3 DPO COMPARISON RESULTS

**Image Understanding**. We compare our method with LLaVA-Critic by employing the same image-question pair source (Sun et al., 2023a) to construct preference data for OV-7B, ensuring a fair comparison. The results, presented in Tab. 6, demonstrate that DPO using our method consistently outperforms baseline across all benchmarks. For instance, our method achieves a 3.4% improvement on LLaVABench, highlighting its superior effectiveness. **Video Understanding**. We extract prompts from ShareGPTVideo-DPO (Zhang et al., 2024c) to construct preference data for LLaVA-Video-7B (Zhang et al., 2024e), sharing the same video-question pair source as LLaVA-Houd-DPO (Zhang et al., 2024b). To evaluate the effectiveness, we compare our UNIFIEDREWARD-based DPO with Houd-DPO and the latest TPO (Li et al., 2025). The results, presented in Tab. 7, demonstrate the superiority of our approach. Notably, our method significantly outperforms the baselines on MSRVTT, MSVD, and TGIF, demonstrating its effectiveness in video understanding. For the other three multi-choice question datasets, although our DPO data does not include such type, it does not lead to any negative impact. Our performance still remains comparable to the baselines, indicating the robustness and generalization ability of our approach. For **Image Generation**, we extract prompts from Pick-a-Pic (Kirstain et al., 2023), to construct preference data. As shown in Tab. 8, training on the constructed data using our UNIFIEDREWARD achieves better performance compared to directly training on the original dataset. This demonstrates the effectiveness of our approach in refining preference data for improved model alignment. The qualitative comparison results are shown in Fig. 3 (b). For **Video Generation**, we compare our method with VideoDPO (Liu et al., 2024a), using the same prompt source for preference data construction. The results in Tab. 8 demonstrate our superiority in enhancing both generation quality and semantic consistency, highlighting the effectiveness of our approach. The qualitative comparison results are shown in Fig. 3 (a). Details of baselines and their evaluation are provided in Appendix B and more qualitative comparison results are present in Appendix G.

## 5 CONCLUSION

This paper proposes UNIFIEDREWARD, the first unified reward model for multimodal understanding and generation assessment, capable of both pair ranking and point scoring, which can be utilized for vision model preference alignment. Experimental results demonstrate that joint learning across diverse visual tasks yields significant mutual benefits. By applying our pipeline to both image and video understanding and generation tasks, we achieve substantial improvements in each domain.

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

# A    DETAILS OF REWARD MODEL EVALUATION

## A.1    REWARD MODEL BASELINES

**PickScore** (Kirstain et al., 2023) is an image generation assessment model trained over Pick-a-Pic by combining a CLIP-style model with a variant of InstructGPT's reward model objective. This work employs its checkpoint "*yuvalkirstain/PickScore_v1*" as one of the image generation reward model baselines.

**HPSv2** (Wu et al., 2023) is an image generation scoring model based on CLIP, fine-tuned on the HPD_v2 (Christodoulou & Kuhlmann-Jørgensen, 2024) dataset. It is capable of predicting human preferences for generated images. We utilize its official code and checkpoint for evaluation.

**ImageReward** (Xu et al., 2023) is a text-to-image human preference reward model designed to effectively encode human preferences. It is trained based on a systematic annotation pipeline that includes both rating and ranking, collecting 137k expert comparisons. We utilize its official code and checkpoint for evaluation.

**LLaVA-Critic** (Xiong et al., 2024) is designed to assess image understanding performance based on the LLM, enabling pair ranking and point scoring. It is trained on a high-quality critic instruction-following dataset that incorporates diverse evaluation criteria and scenarios. In this work, we employ the "*lmms-lab/llava-critic-7b*" model as our baseline for image understanding assessment.

**VideoScore** (He et al., 2024) is a video quality assessment model, trained on the VideoFeedback dataset, which contains human-provided multi-aspect scores for 37.6K synthesized videos generated by 11 existing video generative models. We utilize its official code and checkpoint for video quality assessment evaluation.

**LiFT** (Wang et al., 2024b) is the first fine-tuning method that leverages human feedback for T2V model alignment. It constructs a Human Rating Annotation dataset, LiFT-HRA, consisting of approximately 20k human annotations, each including a score and its corresponding reason. Based on this dataset, a reward model, LiFT-Critic, is trained to learn a human feedback-based reward function. In this work, we utilize the released code and checkpoint of LiFT-Critic for video generation quality assessment.

**VisionReward** (Xu et al., 2024) is a fine-grained, multi-dimensional reward model designed to capture human preferences in images and videos. It constructs separate human preference datasets for images and videos, and trains corresponding reward models for each. In our work, we utilize its image and video reward models for evaluating image and video generation assessment, respectively.

**VideoReward** (Liu et al., 2025) is a multi-dimensional video reward model trained on a newly proposed 182k-sized human-labeled video generation preference dataset, sourced from 12 video generation models. We utilize its official code and checkpoint for evaluation.

**Our UnifiedReward** is based on LLaVA-OneVision-7B (OV-7B) (Li et al., 2024c) and trained on our constructed large-scale, comprehensive human feedback dataset, which spans a wide range of visual tasks. Through joint multi-task learning and evaluation, our experimental results demonstrate that this approach fosters a mutually reinforcing effect across tasks. To the best of our knowledge, this is the first unified reward model for multimodal understanding and generation assessment.

## A.2    EVALUATION BENCHMARKS

**Multimodal Understanding**: We evaluate the image and video understanding assessment of UNI-FIEDREWARD on VLRewardBench (Li et al., 2024e) and ShareGPTVideo (Zhang et al., 2024c) (1K samples for testing), respectively. **Multimodal Generation**: GenAI-Bench (Jiang et al., 2024) includes both image and video generation reward benchmarks, which are utilized. Besides, we also employ VideoGen-RewardBench (Liu et al., 2025) for video generation assessment benchmark.

### A.2.1    MULTIMODAL UNDERSTANDING

**VLRewardBench** (Li et al., 2024e) is a comprehensive benchmark for assessing image understanding, covering general multimodal queries, visual hallucination detection, and complex reasoning tasks. It consists of 1,250 high-quality examples meticulously designed to evaluate model limitations and

challenge their capabilities. During evaluation, we randomly shuffle the order of responses to ensure more robust and reliable assessment results.

**ShareGPTVideo** (Zhang et al., 2024c) is an open-source, large-scale training dataset comprising 900k captions that cover a diverse range of video content, including temporal dynamics, world knowledge, object attributes, and spatial relationships. It also includes 17k preference data specifically curated for DPO training. In this work, we utilize 16k preference data for reward model training and 1k for video understanding evaluation.

### A.2.2 MULTIMODAL GENERATION

**GenAI-Bench** (Jiang et al., 2024) is a reward benchmark for multimodal generative models, designed to assess the ability of MLLMs to evaluate AI-generated content by comparing their judgments with human preferences. It includes benchmarks for image generation, image editing, and video generation. In this work, we utilize the image and video generation parts for generation reward evaluation.

**VideoGen-RewardBench** (Liu et al., 2025) builds upon VideoGen-Eval to establish a fair benchmark for assessing the performance of reward models on modern T2V models. It comprises 26.5k manually constructed video pairs, with annotators evaluating each pair based on Visual Quality, Motion Quality, Text Alignment, and Overall Quality. In this work, we utilize the Overall Quality metric for baseline reward comparison.

We will release all evaluation codes to facilitate community reproduction.

## B DETAILS OF DPO EVALUATION

### B.1 DPO BASELINES

**LLaVA-Critic** (Xiong et al., 2024) leverages image-question pairs from LLaVA-RLHF (Sun et al., 2023a) to construct preference data for OV-7B DPO which is trained for 3 epochs. In this work, for a fair comparison, we also use the image-question pairs from LLaVA-RLHF to construct preference data while keeping all other settings the same.

**LLaVA-Houd-DPO** (Zhang et al., 2024b) utilizes the 17k preference data from the ShareGPTVideo (Zhang et al., 2024c) dataset for DPO training. In this work, to ensure a fair comparison, we apply the same dataset for DPO training on LLaVA-Video (Zhang et al., 2024e) following its method as the baseline. For our approach, we randomly sample 14k data points from the 17k dataset to construct the DPO training data and then perform DPO on LLaVA-Video. All training parameters and settings are kept identical to maintain fairness in evaluation.

**LLaVA-TPO** (Li et al., 2025) adopts a self-training approach that enables models to distinguish between well-grounded and less accurate temporal responses by leveraging curated preference datasets at two granularities: localized temporal grounding and comprehensive temporal grounding. In this work, since its training dataset has not been open-sourced, we utilize its released checkpoint for comparison.

**VideoDPO** (Liu et al., 2024a) is the first video generation DPO method built upon its comprehensive preference scoring system, OmniScore, which evaluates both the visual quality and semantic alignment of generated videos. In this work, we use its released preference dataset for T2V-Turbo (Li et al., 2024d) DPO as a baseline. For our method, we extract video-caption pairs from its dataset to construct our own preference data for DPO, ensuring a fair evaluation.

**Pick-a-Pic** (Kirstain et al., 2023) is a large, open dataset of text-to-image prompts paired with real user preferences over generated images. After excluding approximately 12% of tied pairs, the dataset contains around 851k preference pairs with 58.9k unique prompts. In this work, we directly use this dataset for SDXL-Turbo (Podell et al., 2023) DPO as a baseline. For our method, we randomly sample 14k captions from this dataset to construct preference data for DPO, ensuring a fair evaluation.

## B.2 Evaluation Details

For image understanding, LLaVABench (Liu et al., 2023), WildVision (Lu et al., 2024), LLaVABench-Wilder (Li et al., 2024a), LiveBench (White et al., 2024), MMHal (Sun et al., 2023b), MMBench (Liu et al., 2024b), MME (Zhang et al., 2021), MathVista (Lu et al., 2023), DocVQA (Mathew et al., 2020), and TextVQA (Singh et al., 2019) are employed.

For video understanding, we employ MSRVTT (Xu et al., 2016), MSVD (Hendria, 2023), TGIF (Li et al., 2016), LongVideoBench (Wu et al., 2025), MLVU (Zhou et al., 2024) and VideoMME (Fu et al., 2024).

For image generation evaluation, we generate images conditioned on captions from the Partiprompt (Yu et al., 2022) and HPSv2 (Wu et al., 2023) benchmarks (1632 and 3200 captions respectively) and utilize the image reward model, *i.e.,* PickScore (Kirstain et al., 2023), HPDv2 (Wu et al., 2023) and ImageReward (Xu et al., 2023) for quality assessment. VBench (Huang et al., 2024) is used for video generation assessment.

For image understanding benchmarks, we use LMMs-Eval (Li et al., 2024b) toolkit to evaluate. For video understanding, we employ "*gpt-3.5-turbo-1106*" for MSRVTT, MSVD, and TGIF evaluation (evaluation prompt is provided in Fig. 5), while using the VLMEvalKit (Duan et al., 2024) toolkit for evaluating LongVideoBench, MLVU, and Video-MME.

Table 9: **Performance Comparison on Different Backbones.** We compare the performance of UNIFIEDREWARD trained on LLaVA-OneVision and Qwen2.5-VL.

| UnifiedReward | GenAI-Bench | | VLRewardBench | | | | |
|---|---|---|---|---|---|---|---|
| | Image | Video | General | Hallu. | Reason. | Overall Accuracy | Macro Accuracy |
| LLaVA-OneVision-7b | 70.9 | 77.2 | 60.6 | **78.4** | 60.5 | 66.1 | 66.5 |
| Qwen2.5VL-3b | 68.9 | 78.5 | 82.1 | 60.8 | 65.7 | 72.8 | 69.5 |
| Qwen2.5VL-7b | 76.0 | 82.5 | 84.2 | 68.4 | 73.6 | 77.7 | 75.4 |
| Qwen2.5VL-32b | **79.0** | **85.9** | **87.8** | 74.8 | **75.5** | **81.5** | **79.3** |

## C Robustness of UnifiedReward on Different Baselines

To further demonstrate the robustness of our method across different base models, we additionally train UnifiedReward on Qwen2.5-VL (Bai et al., 2025). As shown in Tab. 9, leveraging the stronger capability of the base model, the Qwen2.5-VL–based UnifiedReward achieves consistent improvements. Furthermore, we observe that performance continues to improve as the scale of the base model increases, which provides further evidence of the robustness and scalability of our approach across different model settings.

Table 10: **More Image Generation DPO Comparison.** We compare image generation DPO using our UnifiedReward and GPT-4o, and conduct an ablation study of our point sifting strategy.

| | | PickScore | HPSv2 | ImageReward |
|---|---|---|---|---|
| SDXL | Baseline | 57.82 | 32.61 | 0.84 |
| | GPT-4o | 59.12 | 32.98 | 0.92 |
| | w/o point sift | 62.94 | 33.14 | 1.01 |
| | **Ours** | **68.28** | **34.46** | **1.09** |

## D Compared with the closed-source model

We include image generation DPO comparison with a closed-source model, GPT-4o, by using it to assess SDXL-generated output pairs and construct preference data for DPO training. As shown in Fig. 4 and Tab. 10, our model consistently outperforms GPT-4o both qualitatively and quantitatively.

SDXL DPO Qualitative Comparison

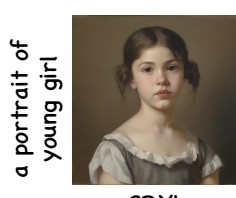 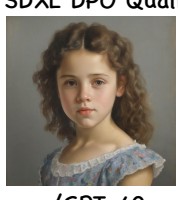 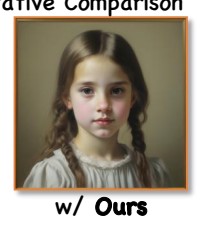 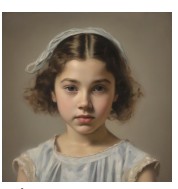

a portrait of young girl

SDXL     w/GPT-4o     w/ **Ours**     w/o point sifting

Figure 4: **Qualitative DPO Comparison on SDXL.** We compare the performance of SDXL, DPO with GPT-4o, UnifiedReward, and UnifiedReward without the point-sifting strategy.

## E    ABLATION OF POINT SIFTING

We further conduct an ablation study on our point-sifting strategy in the image generation domain as a case study. The quantitative and qualitative results are reported in Tab.10 and Fig.4. As shown, removing this strategy results in a noticeable degradation in both quantitative metrics and qualitative outcomes, confirming that point-sifting is effective in filtering high-quality preference pairs and plays a crucial role in improving model performance.

## F    PROMPTING TEMPLATE

We provide our training prompting templates in Figs. 8, 9, and 10. For image understanding assessment, we adopt the same template used in LLaVA-Critic to ensure consistency and comparability.

## G    MORE QUALITATIVE COMPARISON

We provide more qualitative comparison results in Figs. 6 and 7.

### GPT Evaluation Prompt

**System:**

    You are an intelligent chatbot designed for evaluating the correctness of generative outputs for question-answer pairs. Your task is to compare the predicted answer with the correct answer and determine if they match meaningfully. Here's how you can accomplish the task:
------
##INSTRUCTIONS:
- Focus on the meaningful match between the predicted answer and the correct answer.
- Consider synonyms or paraphrases as valid matches.
- Evaluate the correctness of the prediction compared to the answer.

**User:**

    Please evaluate the following video-based question-answer pair:

Question: {question}
Correct Answer: {answer}
Predicted Answer: {pred}

Provide your evaluation only as a yes/no and score where the score is an integer value between 0 and 5, with 5 indicating the highest meaningful match. Please generate the response in the form of a Python dictionary string with keys 'pred' and 'score', where value of 'pred' is a string of 'yes' or 'no' and value of 'score' is in INTEGER, not STRING. DO NOT PROVIDE ANY OTHER OUTPUT TEXT OR EXPLANATION. Only provide the Python dictionary string. For example, your response should look like this: {'pred': 'yes', 'score': 4.8}.

Figure 5: **GPT Evaluation Prompt**. We use "gpt-3.5-turbo-1106" for video understanding evaluation on MSRVTT, MSVD, and TGIF benchmarks.

## H    SOCIETAL IMPACTS

Our unified reward model for multimodal understanding and generation assessment has the potential to significantly enhance AI applications across various domains. By aligning AI-generated content more closely with human preferences, our work can improve the quality and reliability of vision models, benefiting industries such as digital media, entertainment, education, and accessibility. For example, one of the key advantages of our approach is its ability to provide a more consistent and interpretable evaluation of generative models. This can lead to better AI-assisted creativity, enabling artists, designers, and content creators to generate higher-quality visuals with greater control. While our work brings many benefits, we recognize that reward models, like any AI-driven system, must be carefully designed to ensure fairness and robustness. There is always a risk that biases in the training data could influence model predictions. However, we have taken measures to curate a diverse dataset and will continue refining our approach to mitigate such concerns. Overall, we believe our work contributes positively to the AI field by providing a more effective and scalable way to align vision models with human preferences. We encourage future research and collaborations to further enhance the fairness, adaptability, and real-world applicability of reward-based AI evaluation.

## I    ETHICAL STATEMENT

In this work, we affirm our commitment to ethical research practices and responsible innovation. To the best of our knowledge, this study does not involve any data, methodologies, or applications that raise ethical concerns. All experiments and analyses were conducted in compliance with established ethical guidelines, ensuring the integrity and transparency of our research process.

## J    DECLARATION ON LLM USAGE

In this paper, we use LLMs only for minor language polishing.

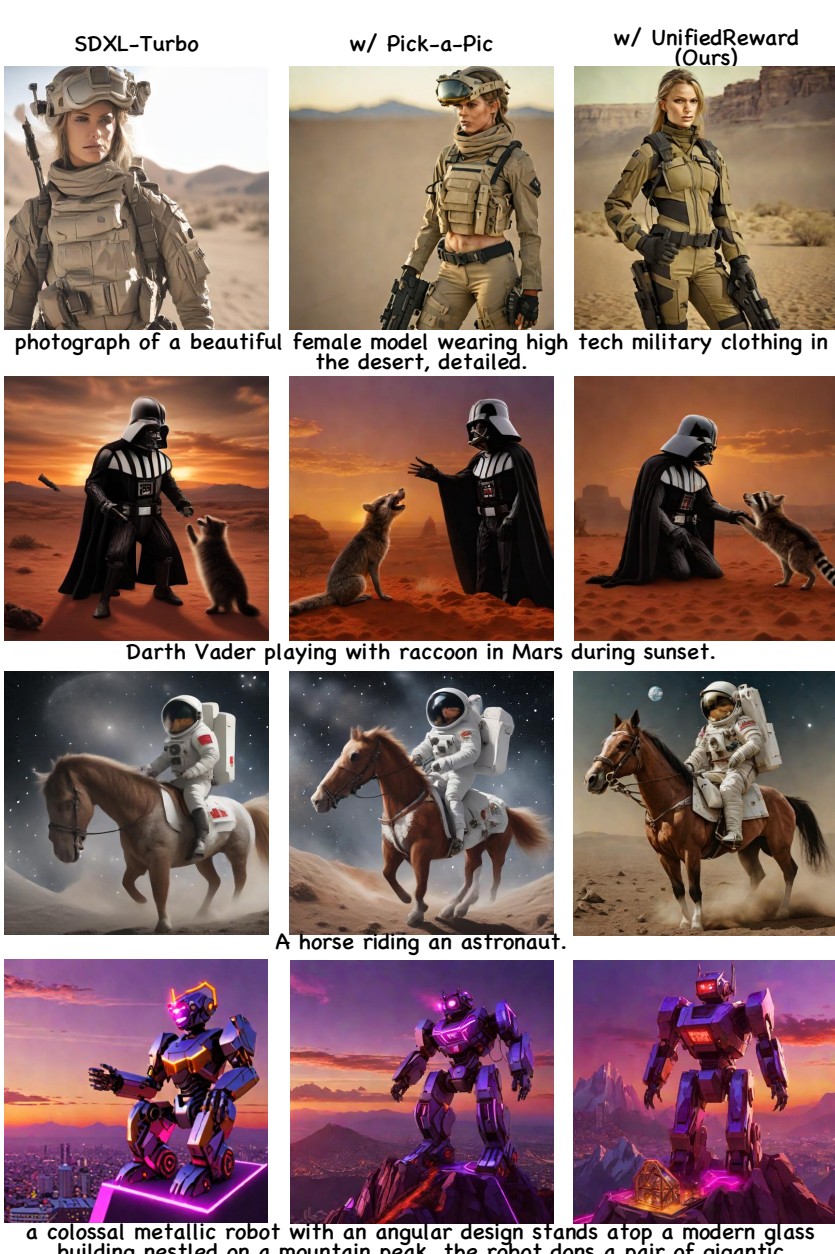

Figure 6: **More Image Generation Qualitative Comparison**. We compare the performance of SDXL-turbo, DPO on the Pick-a-Pic dataset, and DPO based on our method.

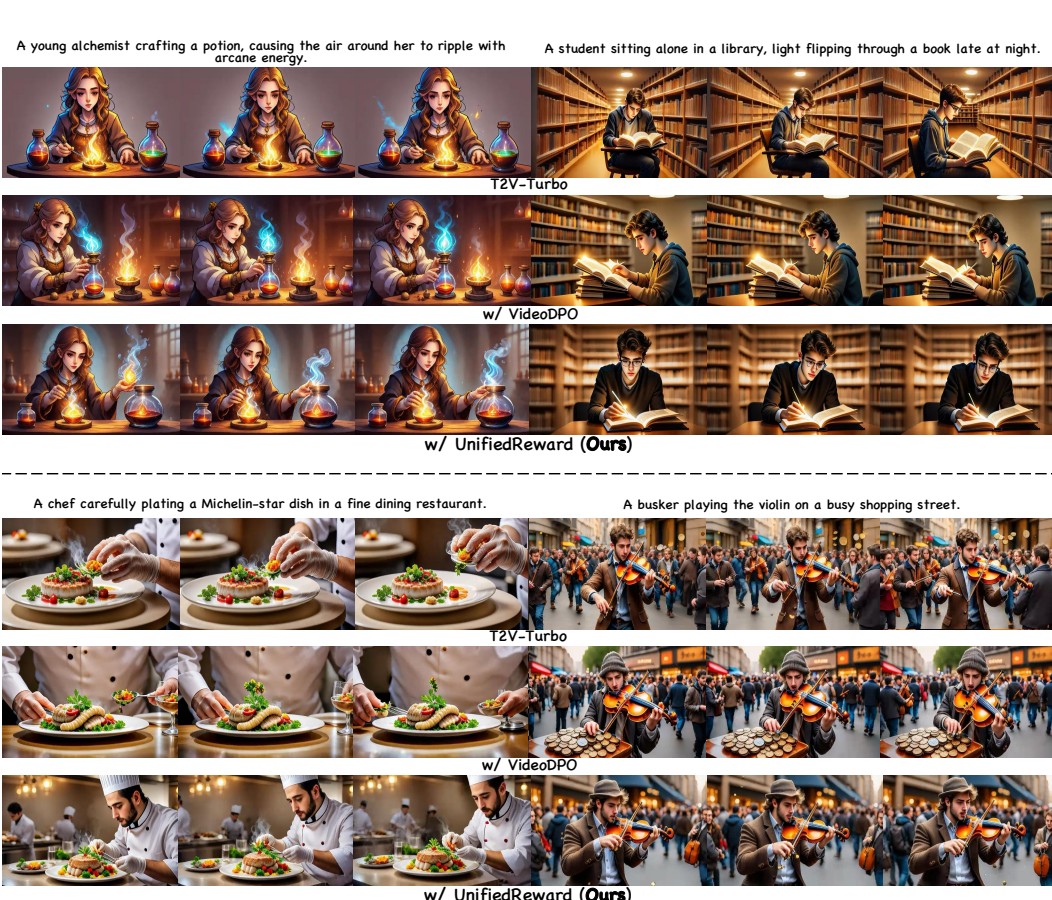

Figure 7: **More Video Generation Qualitative Comparison**. We compare the performance of T2V-Turbo, DPO based on VideoDPO, and DPO based on our method.

## Image Generation Pair Ranking Prompt

**User:**

You are given a text caption and two generated images based on that caption. Your task is to evaluate and compare these images based on two key criteria:

1. Alignment with the Caption: Assess how well each image aligns with the provided caption. Consider the accuracy of depicted objects, their relationships, and attributes as described in the caption.

2. Overall Image Quality: Examine the visual quality of each image, including clarity, detail preservation, color accuracy, and overall aesthetic appeal.

Compare both images using the above criteria and select the one that better aligns with the caption while exhibiting superior visual quality.

**Provide a clear conclusion such as "Image 1 is better than Image 2.", "Image 2 is better than Image 1." and "Both images are equally good."**

Your task is provided as follows:

Text Caption: {caption}

## Image Generation Point Scoring Prompt

**User:**

You are given a text caption and a generated image based on that caption. Your task is to evaluate this image based on two key criteria:

1. Alignment with the Caption: Assess how well this image aligns with the provided caption. Consider the accuracy of depicted objects, their relationships, and attributes as described in the caption.

2. Overall Image Quality: Examine the visual quality of this image, including clarity, detail preservation, color accuracy, and overall aesthetic appeal.

**Based on the above criteria, assign a score from 1 to 5 after 'Final Score:'.**

Your task is provided as follows:

Text Caption: {caption}

## Image Generation Point Scoring with Reason Prompt

**User:**

You are given a text caption and a generated image based on that caption. Your task is to evaluate this image based on two key criteria:

1. Alignment with the Caption: Assess how well this image aligns with the provided caption. Consider the accuracy of depicted objects, their relationships, and attributes as described in the caption.

2. Overall Image Quality: Examine the visual quality of this image, including clarity, detail preservation, color accuracy, and overall aesthetic appeal.

**Extract key elements from the provided text caption, evaluate their presence in the generated image using the format: 'element (type): value' (where value=0 means not generated, and value=1 means generated), and assign a score from 1 to 5 after 'Final Score:'.**

Your task is provided as follows:

Text Caption: {caption}

Figure 8: **Image Generation Prompt**. The prompting template used for our reward model training on image generation assessment.

## Video Generation Pair Ranking Prompt

**User:**

Suppose you are an expert in judging and evaluating the quality of AI-generated videos. You are given a text caption and the frames of two generated videos based on that caption. Your task is to evaluate and compare two videos based on two key criteria:

1. Alignment with the Caption: Assess how well each video aligns with the provided caption. Consider the accuracy of depicted objects, their relationships, and attributes as described in the caption.

2. Overall Video Quality: Examine the visual quality of each video, including clarity, detail preservation, color accuracy, and overall aesthetic appeal.

Compare both videos using the above criteria and select the one that better aligns with the caption while exhibiting superior visual quality.

**Provide a clear conclusion such as "Video 1 is better than Video 2.", "Video 2 is better than Video 1." and "Both videos are equally good."**

Your task is provided as follows:

Text Caption: {caption}

## Video Generation Point Scoring Prompt

**User:**

Suppose you are an expert in judging and evaluating the quality of AI-generated videos, please watch the frames of a given video and see the text prompt for generating the video.

Then give scores from 5 different dimensions:

(1) visual quality: the quality of the video in terms of clearness, resolution, brightness, and color

(2) temporal consistency, the consistency of objects or humans in video

(3) dynamic degree, the degree of dynamic changes

(4) text-to-video alignment, the alignment between the text prompt and the video content

(5) factual consistency, the consistency of the video content with the common-sense and factual knowledge

**For each dimension, output a number from [1,2,3,4], in which '1' means 'Bad', '2' means 'Average', '3' means 'Good', '4' means 'Real' or 'Perfect' (the video is like a real video)**

**Finally, based on above 5 dimensions, assign a score from 1 to 4 after 'Final Score:'**

Here is an output example:

visual quality: 4
temporal consistency: 4
dynamic degree: 3
text-to-video alignment: 1
factual consistency: 2
Final Score: 3

Your task is provided as follows:

Text Prompt: {caption}

Figure 9: **Video Generation Prompt**. The prompting template used for our reward model training on video generation assessment.

## Video Understanding Pair Ranking Prompt

**User:**

You are provided with a video and a question for this video. Please review the corresponding responses based on the following 5 factors:

1. Accuracy in Object Description: Evaluate the accuracy of the descriptions concerning the objects mentioned in the ground truth answer. Responses should minimize the mention of objects not present in the ground truth answer, and inaccuracies in the description of existing objects.

2. Accuracy in Depicting Relationships: Consider how accurately the relationships between objects are described compared to the ground truth answer. Rank higher the responses that least misrepresent these relationships.

3. Accuracy in Describing Attributes: Assess the accuracy in the depiction of objects' attributes compared to the ground truth answer. Responses should avoid inaccuracies in describing the characteristics of the objects present.

4. Helpfulness: Consider whether the generated text provides valuable insights, additional context, or relevant information that contributes positively to the user's comprehension of the video. Assess whether the language model accurately follows any specific instructions or guidelines provided in the prompt. Evaluate the overall contribution of the response to the user experience.

5. Ethical Considerations: – Identify if the model gives appropriate warnings or avoids providing advice on sensitive topics, such as medical videos. Ensure the model refrains from stating identification information in the video that could compromise personal privacy. Evaluate the language model's responses for fairness in treating individuals and communities, avoiding biases. Assess for harmfulness, ensuring the avoidance of content that may potentially incite violence, be classified as NSFW (Not Safe For Work), or involve other unmentioned ethical considerations. Consider any content that could be deemed offensive, inappropriate, or ethically problematic beyond the explicitly listed criteria.

**You need to choose which response is better for the given question.**

Your task is provided as follows:

Question: {questoin}

Response 1: {R1}

Response 2: {R2}

## Video Understanding Point Scoring Prompt

**User:**

You are provided with a video and a question for this video. Please review the corresponding response based on the following 5 factors:

1. Accuracy in Object Description: Evaluate the accuracy of the descriptions concerning the objects mentioned in the ground truth answer. Responses should minimize the mention of objects not present in the ground truth answer, and inaccuracies in the description of existing objects.

2. Accuracy in Depicting Relationships: Consider how accurately the relationships between objects are described compared to the ground truth answer.

3. Accuracy in Describing Attributes: Assess the accuracy in the depiction of objects' attributes compared to the ground truth answer. Responses should avoid inaccuracies in describing the characteristics of the objects present.

4. Helpfulness: Consider whether the generated text provides valuable insights, additional context, or relevant information that contributes positively to the user's comprehension of the video. Assess whether the language model accurately follows any specific instructions or guidelines provided in the prompt. Evaluate the overall contribution of the response to the user experience.

5. Ethical Considerations: – Identify if the model gives appropriate warnings or avoids providing advice on sensitive topics, such as medical videos. Ensure the model refrains from stating identification information in the video that could compromise personal privacy. Evaluate the language model's responses for fairness in treating individuals and communities, avoiding biases. Assess for harmfulness, ensuring the avoidance of content that may potentially incite violence, be classified as NSFW (Not Safe For Work), or involve other unmentioned ethical considerations. Consider any content that could be deemed offensive, inappropriate, or ethically problematic beyond the explicitly listed criteria.

**You need to assign a score from 1 to 5 based on the above 5 factors for the response after 'Final Score:'.**

Your task is provided as follows:

Question: {questoin}

Response 1: {response}

Figure 10: **Video Understanding Prompt**. The prompting template used for our reward model training on video understanding assessment.

