# OpenReview forum: "Unified Reward Model for Multimodal Understanding and Generation"
_ICLR.cc/2026/Conference — ICLR 2026 Conference Withdrawn Submission_

### Official Review · Reviewer_6Wnv · 2025-10-30

**Soundness:** 3
**Presentation:** 3
**Contribution:** 3
**Rating:** 6
**Confidence:** 4

**Summary:**

This paper proposes UNIFIEDREWARD, the first unified reward model for multimodal understanding and generation assessment, addressing the limitation that existing reward models are task-specific. This work assembles a large-scale unified preference dataset covering image/video understanding and generation tasks, demonstrating significant advantages in both generation and understanding tasks.

**Strengths:**

1. This work integrates image and video generation and understanding tasks to construct a universal reward model, demonstrating effectiveness through joint training on an integrated large-scale dataset.

2. The model shows notable performance advantages in both generation and understanding tasks.

**Weaknesses:**

1. The paper claims that joint training of multiple tasks has a synergistic effect, but could it be possible that the performance improvement is simply due to the larger amount of data when training multiple tasks together compared to single task, rather than true synergy? Could the authors conduct an ablation study by reducing the joint training dataset size to match that of single-task dataset to verify the validity of this claim?

2. Could the base models used to generate positive and negative samples during DPO optimization be stronger than the baseline datasets being compared? For example, is the base model used for image generation of higher quality than the Pick-a-Pic dataset? If such unfairness exists, could the authors use PickScore to score the new dataset and construct preference pairs for a fairer comparison?

3. Why not directly use the reward model for reinforcement learning? Reconstructing a DPO dataset seems quite time-consuming.

**Questions:**

Responses to the questions raised in the weakness section:

---

### Official Review · Reviewer_8PQb · 2025-10-30

**Soundness:** 3
**Presentation:** 3
**Contribution:** 2
**Rating:** 4
**Confidence:** 4

**Summary:**

This paper proposes UnifiedReward, the first unified reward model that jointly learns to assess both image and video understanding/generation tasks. The motivation is that current reward models are task-specific and lack generalizability across different modalities and objectives. To deal with it, the authors construct a large-scale human preference dataset containing 236K samples, which combines multiple existing datasets. A vision-language model (LLaVA-OneVision-7B) is fine-tuned on this data, so as to predict both pairwise rankings and pointwise scores. This model is then used to generate high-quality preference data to fine-tune downstream vision and diffusion models using DPO. Experiments on a wide range of benchmarks show that joint multi-task training brings cross-domain benefits, improving performance across both understanding and generation tasks, including image and video domains.

**Strengths:**

1.	The paper tries to resolve an important problem—generalizing reward models across multiple visual modalities and tasks. The unified perspective is valuable and enlightening. The proposed method can asses both image and video understanding and generation.
2.	The authors combine numerous high-quality human preference datasets and preprocess them into a consistent unified framework, covering both pairwise ranking and pointwise scoring setups, which is an engineering and data contribution.
3.	The presentation of methodology is clear and systematic. A well-structured training and application pipeline is
4.	Extensive experiments across multiple tasks and modalities show consistent improvements. The multi-task analysis supports the central hypothesis that cross-task joint learning improves generalization.

**Weaknesses:**

1.	The paper did not discuss why multitasking training brings complementary gains. The performance gains may come from more training data rather than multi-modal and multi-task learning.
2.	Although the paper’s idea is great, the methodology lacks novelty. The proposed UnifiedReward mainly integrates existing components, including LLaVA, pairwise/pointwise cross-entropy and DPO. It seems more like an engineering improvement.
3.	The paper does not clearly explain how different datasets are integrated. Besides, the unified dataset shows data imbalance. For example, Fig. 1(b) indicates that the pairwise data for video generation is far less than pointwise tasks. No evidence or experiments shows that the synergistic effect of multitask learning can alleviate this deficiency. The deficiency of the unified dataset may impair the performance of UnifiedReward model.

**Questions:**

1.	How can you ensure the performance gains come from task synergy rather than simply having more diverse training data? Could you provide more experimental or visual evidence to support that the reward embeddings has truly learned a cross-modal shared semantic space?
2.	How sensitive is the joint training to data distribution imbalance? Does the imbalance affects the performance of the reward model? What happens if you balance the dataset?

---

### Official Review · Reviewer_sB6a · 2025-10-31

**Soundness:** 2
**Presentation:** 2
**Contribution:** 2
**Rating:** 4
**Confidence:** 3

**Summary:**

This paper introduces UNIFIEDREWARD, the first unified reward model designed to assess both multimodal understanding (e.g., video Q&A) and generation (e.g., text-to-image) tasks across both image and video modalities.

**Strengths:**

Pros:
1. The concept of a single, unified reward model for such a broad range of visual tasks is a good contribution. The paper effectively argues for the limitations of task-specific models and presents a compelling hypothesis about cross-task synergy.
2. The "pair ranking + point sifting" strategy for data construction is an interesting approach. It combines the strengths of relative comparison (ranking) and absolute quality assessment (scoring) to filter for the most informative preference pairs, likely contributing significantly to the DPO performance.
3.  Instead of undertaking the massive cost of new human annotation, the authors pragmatically curated and standardized a large dataset from multiple existing public sources.

**Weaknesses:**

Cons:
1. This paper could benefit from a deeper analysis of the resulting unified dataset. Questions remain about potential overlaps, domain biases, or conflicting preference signals. The paper assumes synergy is always positive, but it would be valuable to discuss the potential for negative transfer (e.g., if aesthetic generation criteria conflict with factual understanding criteria).
2. The "point sifting" step is described as selecting the chosen output with the maximum score and the rejected output with the minimum score. This could be brittle. For instance, if all generated outputs are poor, the "best" one might still be bad. More detail on whether thresholds are used or how low-quality pairs are handled would strengthen the methodology.
3. The paper mentions that for pointwise scoring, they "do not enforce a unified response format or score range" (lines 230-231). This is a potential source of instability. It is unclear how the model effectively learns to produce meaningful scores when its training targets come from different scales (e.g., 1-5, 1-10, etc.) and how these heterogeneous scores are reliably compared during the "point sifting" phase.
4. More details on datasets are desired. During the training of UNIFIEDREWARD, how were the different tasks and modalities balanced? Given the varying sizes of the source datasets (as seen in Figure 1b), was a specific sampling strategy employed to prevent the model from being biased towards tasks with more data, such as image understanding?
5. In the DPO preference data construction pipeline, a vision model generates N candidate outputs. What was the value of N used in the experiments? This number is critical for understanding the computational cost and the filtering ratio of the data curation process.
6. Did the authors observe any cases where the model struggled to balance conflicting objectives? For example, in an image understanding task, a response might be factually correct but poorly worded, while another is eloquent but slightly inaccurate. How does a unified model learn to navigate such trade-offs between correctness, aesthetics, and helpfulness?
7. The paper presents strong quantitative results and qualitative successes. A qualitative analysis of failure cases for UNIFIEDREWARD would be highly informative. When does it make mistakes? Does it have specific blind spots, for example, in evaluating complex reasoning or subtle artistic styles?

**Questions:**

See Weaknesses

---

### Official Review · Reviewer_hLSF · 2025-11-03

**Soundness:** 2
**Presentation:** 1
**Contribution:** 2
**Rating:** 2
**Confidence:** 4

**Summary:**

This paper proposes UNIFIEDREWARD, a single reward model to evaluate and guide image + video understanding and generation tasks simultaneously. The method aggregates multiple existing human-preference datasets, trains a unified reward model based on OV-7B, and applies it to construct preference pairs for DPO of VLMs and diffusion models. Experiments report consistent performance gains on several benchmarks such as VLRewardBench, GenAI-Bench, and VBench.

**Strengths:**

- The idea of unifying reward modeling across understanding and generation tasks is timely.

- The authors aggregate many datasets and evaluate across multiple benchmarks (image/video understanding and generation).

- Figures 1–2 clearly depict the data-construction and training pipelines.

**Weaknesses:**

- The claimed “unification” is essentially a joint training scheme over concatenated datasets, rather than a novel learning principle or architectural innovation, and thus offers limited insight to the community.

- Most “new” preference data are re-labeled or merged from public sources (EvalMuse, HPD, LLaVA-Critic). The authors provide no re-annotation quality control.

- Section 3.4 gives equations copied almost verbatim from [1] and [2]. There is no explanation of how the same DPO loss is simultaneously applied to diffusion and language-based decoders.

- Many metrics (e.g., “Macro Accuracy” “Overall Accuracy”) lack definition.

- The paper contains near-verbatim repetition between Section 3.2 (“Unified Preference Dataset Construction” lines 110–150) and Section 3.3 (“Unified Embedding Alignment” lines 160–200). Both sections restate almost identical sentences about dataset merging, unified schema, and shared embedding space (e.g., “To effectively train a unified reward model…” and “To bridge the gap between understanding and generation modalities…”). The second occurrence provides no new information, suggesting careless editing and weakening the overall presentation quality.

- Some references (Sun et al., 2023a/b/c) are copy-pasted thrice.

**References**

[1] Diffusion Model Alignment Using Direct Preference Optimization (Wallace et al., 2024)

[2] LiFT: Leveraging Human Feedback for Text‑to‑Video Model Alignment (Wang et al., 2024)

**Questions:**

n/a

---

### Note · Authors · 2025-11-14

I have read and agree with the venue's withdrawal policy on behalf of myself and my co-authors.